A study of diel and seasonal patterns of loss of commercial lychee fruits to vertebrate frugivores: implications for mitigating a human-wildlife conflict

Bhanda Geetika geetika.bhanda1@umail.uom.ac.mu 1
Oleksy Ryszard Z. 2
Reinegger Raphaël D. 3
Baider Cláudia 4
Florens F.B. Vincent 1
1 Tropical Island Biodiversity, Ecology and Conservation Pole of Research, Faculty of Science, University of Mauritius , Le Réduit , Mauritius
2 Ecosystem Restoration Alliance Indian Ocean , St. Pierre , Mauritius
3 School of Biological Sciences, University of Bristol , Bristol , United Kingdom
4 The Mauritius Herbarium, RE Vaughan Building, Agricultural Services, Ministry of Agro-Industry, Food Security, Blue Economy and Fisheries , Le Réduit , Mauritius
Kramer Donald
Electronic publication date: 2025 Apr 21
Publication date: 2025
Volume: 13
Electronic Location ID: e19269
Received 2024 Dec 18; Accepted 2025 Mar 14
Copyright: ©2025 Bhanda et al.
Copyright year: 2025
Copyright holder: Bhanda et al.
License: This is an open access article distributed under the terms of the Creative Commons Attribution License, which permits unrestricted use, distribution, reproduction and adaptation in any medium and for any purpose provided that it is properly attributed. For attribution, the original author(s), title, publication source (PeerJ) and either DOI or URL of the article must be cited.
License URL: https://creativecommons.org/licenses/by/4.0/

Keywords: Flying fox, Foraging pattern, Fruit protection, Litchi chinensis, Mauritius, Pteropus niger

Funding: Fondation Segré Conservation Action Fund through IUCN Save Our Species No. 2022C-8 The Agence Française de Développement through the Varuna Biodiversité programme (Project 22-SB3004) The contribution of the late Mr Patrick Alexander towards the support of the registration fees of Ms Bhanda’s doctoral programme for 2024 This work was supported by the Fondation Segré Conservation Action Fund through IUCN Save Our Species (No. 2022C-8), the Agence Française de Développement through the Varuna Biodiversité programme (Project 22-SB3004) and by the contribution of the late Mr Patrick Alexander towards the support of the registration fees of Ms Bhanda’s doctoral programme for 2024. The funders had no role in study design, data collection and analysis, decision to publish, or preparation of the manuscript.

==============================
Human-wildlife conflicts pose a growing threat to biodiversity, particularly when the targeted species plays an ecological keystone role. Mauritius has repeatedly mass-culled an endemic and threatened flying fox species (the Mauritian flying fox; Pteropus niger) failing the intended objectives of crop protection and elevating the species’ extinction risks. In this context, the ecology of this species should be better understood to develop non-lethal management strategies. Here we investigated foraging patterns of vertebrate frugivores over 24 hour cycles in lychee orchards and backyard gardens. We assessed all agents of damage (mainly flying fox, alien bird, alien mammal) and the temporal variation of flying fox and bird foraging (take and amount eaten relative to fruit ripeness) on lychee trees. The most important frugivores foraging on lychees were flying foxes (78.3%) and birds (16.1%), namely ring-necked parakeets (Alexandrinus krameri), red-whiskered bulbuls (Pycnonotus jocosus), village weavers (Ploceus cucullatus) and common mynas (Acridotheres tristis) while damage by alien mammals was negligible (<1%). Flying foxes consumed more fruits in the early night (59%) compared to the late night and this was statistically significant in one orchard and backyards. However, the difference in damage was on average one to three fruits per tree per night. Bird damage at both orchards was highest during the first half of the day (64%). Flying foxes ate fewer fruits towards the end of the fruiting season while birds followed the opposite trend. As fruit ripeness increased from unripe to fully ripe, flying foxes ate 39–42% more lychee pulp per fruit at the two orchards. Parakeets ate 7% more fruit pulp with increasing ripeness at one orchard only. Deliberate disturbances involving smoke, noise or light to deter flying foxes were common in orchards. The weak difference in the extent of flying fox damage to fruits between early and late night suggested at best minor advantages of concentrating deliberate disturbances in early night, and that netting would be a better strategy as it would also protect against diurnal frugivores. Additionally, trees should be protected from the sixth week after fruit set as most damage occurred when fruits were unripe. Such an improved timing of crop protection should play an important role in reducing fruit losses and thereby alleviate the human-wildlife conflict around the flying fox’s diet.

Introduction

Human-wildlife conflict (HWC) arises when human goals are negatively impacted by the needs and behaviour of wildlife or vice versa (Madden, 2004), for instance, when wildlife is posing a threat to human safety or damaging crops leading to human retaliation (Warne & Jones, 2003; Peterson et al., 2010; Florens, 2016). Such conflicts are considered as a serious and growing challenge faced by wildlife (Frank, Glikman & Marchini, 2019). These conflicts, especially in anthropogenic landscapes like agricultural areas (König et al., 2020), are expected to worsen globally (Dickman, 2010; Seoraj-Pillai & Pillay, 2016) as human population size, consumption rates and rates of habitat destruction progress (Ripple et al., 2017; Ripple et al., 2020; Ripple et al., 2021). This is particularly problematic when the species at the centre of the conflict plays an ecological keystone role (Florens et al., 2017), especially where other species playing similar roles have been driven extinct by human activities (Cheke & Hume, 2008) and where species introduced by humans cannot fulfil that role adequately (Heinen et al., 2023). Hence, to address these conflicts, there is an increasing need for novel multi-disciplinary strategies (White & Ward, 2010), focusing on evidence-based approaches (e.g., Florens & Baider, 2019; Siex & Struhsaker, 1999).

Flying foxes (Chiroptera: Pteropodidae) are mainly frugivorous bats that often forage on commercial crops (Aziz et al., 2016) leading to HWC and frequently resulting in persecution, illegal killing (Kingston, Florens & Vincenot, 2023) as well as government-led or permitted culling (Bumrungsri et al., 2009; Epstein et al., 2009; Florens, 2016; Vincenot, Florens & Kingston, 2017). While non-lethal alternatives to protect fruit trees such as netting (Korine, Izhaki & Arad, 1999; Oleksy et al., 2021) or using deterrent systems (Ullio, 2002; Chakravarthy & Girish, 2003) are common, certain difficulties may impede their full implementation such as high costs, labour intensiveness and time they take to be implemented (Gough, 2002; Ullio, 2002; Tollington et al., 2019). In this context, an improved understanding of the bat’s foraging ecology like peak foraging time and foraging behaviour related to human presence and movement ecology is crucial to better devise and optimise non-lethal alternatives that could potentially alleviate HWC (Verghese, 1998; Srinivasulu & Srinivasulu, 2002; Hengjan et al., 2018; Schloesing et al., 2020).

Mauritius appeared in the HWC literature when it started planning mass-culling a species threatened with extinction (Florens, 2012)—the Mauritian flying fox (Pteropus niger)—because it includes commercial fruits like lychee (Litchi chinensis Sonn., Sapindaceae) and mango (Mangifera indica L., Anacardiaceae) in its diet (Tollington et al., 2019; Oleksy et al., 2021). Pteropus niger is a Mascarene (Mauritius, Rodrigues and Réunion islands) endemic (Cheke & Hume, 2008) ecological keystone species (Florens et al., 2017), a role amplified due to human-driven extinctions of other frugivores (Albert et al., 2021; Heinen et al., 2023). It was already threatened with extinction before the mass-culling campaigns (Hutson & Racey, 2013). As fruit growers’ claims of damage to unprotected commercial trees increased from 30–40% in 2012, 73% in 2014 to 75–100% in 2015 (Government of Mauritius, 2010; Anonymous, 2013; Anonymous, 2015a; Anonymous, 2015b), the government weakened the country’s main biodiversity protection law in November 2015 to enable mass-culling of flying foxes (Florens, 2015; Florens, 2016). Repeated mass-culling campaigns (Olival, 2016; Vincenot, Florens & Kingston, 2017; Florens & Vincenot, 2018; Chelvan, 2020) ensued and contributed to the worsening of P. niger’s International Union for Conservation of Nature (IUCN) Red List category (from Vulnerable to Endangered) (Kingston et al., 2018). As scientists predicted, mass-culling was ineffective in increasing fruit production (Olival, 2016; Florens & Baider, 2019).

In HWC such as in Mauritius, the ‘accused’ is typically not the only agent of damage and it is important to distinguish between damage by different frugivores to mitigate HWC. For instance in 1992, Zanzibar red colobus (Piliocolobus kirkii) was perceived as a threat by coconut farmers and after quantifying damage, it was found that damage by sykes monkeys (Cercopithecus mitis albogularis) could have been wrongly blamed on colobus (Siex & Struhsaker, 1999). While flying foxes are the only native frugivores foraging in orchards in Mauritius, several introduced alien species are also known to damage commercial fruits. Reported ones are birds (red-whiskered bulbuls (Pycnonotus jocosus); common mynas (Acridotheres tristis); ring-necked parakeets (Alexandrinus krameri), accounting for 2–12% damage and rats (Rattus spp.), responsible for <1% damage in lychee orchards (Oleksy et al., 2021). Long-tailed macaques (Macaca fascicularis) have also been reported to extensively damage crops on backyard lychee trees, but this could not be confirmed (Tollington et al., 2019). Properly assessing frugivore damage in orchards is the first step in enabling suitable mitigation strategies.

The mitigation or resolution of HWC also depends in part on a good understanding of the temporal expressions of the undesired effects. Both flying foxes and parakeets may leave triangular bite marks when foraging on fruits (Banack, 1998; Senar et al., 2016). Hence, assessing diel patterns of frugivore damage in orchards helps to differentiate between nocturnal and diurnal species. Studies on the diel and seasonal patterns of vertebrate frugivores have also suggested time-specific mitigation strategies. For instance, crop foraging by baboons in South Africa appears to occur mostly before 15:00 and it was proposed that guarding efforts should be focused in the morning and early afternoon for increased effectiveness while monitoring for any temporal change in crop raiding (Walton, Findlay & Hill, 2021). Telemetric data on the foraging activity of flying foxes in Mauritius showed that in regions with commercial fruit trees, their foraging was more concentrated during early nights. Hence, to better focus efforts on protecting orchards, a temporally targeted (18:00–00:00) approach to the use of active deterrence methods (sound, smoke, light) was recommended (Seegobin, Oleksy & Florens, 2022). Bird damage in a vineyard in Canada started early in the ripening season, when fruits were still unripe, and it was recommended that deterrent programmes should be focused when crops were more susceptible to damage (Somers & Morris, 2002). Previous studies in Mauritius also found that flying foxes ate mostly ripe fruits in the forest (Krivek et al., 2020; Reinegger et al., 2021).

Consequently, we investigated the foraging patterns of vertebrate frugivores over 24 h cycles in lychee orchards and backyard gardens in Mauritius. Firstly, we assessed the contribution of different species to fruit damage. Secondly, we evaluated diel and seasonal patterns in vertebrate frugivores’ foraging behaviour. We hypothesised that volant vertebrate frugivores require high energy foods (like lychee) early during foraging to replenish energy levels after their sleep cycle (Bednekoff & Houston, 1994; Srinivasulu & Srinivasulu, 2002). Consequently, we predicted that flying foxes would eat more lychee before than after midnight. Similarly, we predicted that frugivorous birds would eat more fruits in the morning than during other time periods. As fruit abundance is an important cue for fruit selection by frugivores (Manasse & Howe, 1983), we also hypothesised that foraging intensity would increase with fruit availability. Consequently, we predicted that fruit damaged by frugivores would decrease towards the end of the fruiting season when trees have few fruits left. Since some variations exist in the ripeness stages of lychee fruits between trees within an orchard and within the tree, we finally explored the relationship between the amount of fruit pulp consumed per fruit by vertebrate frugivores and its corresponding ripeness stage. As ripe fruits are generally more palatable to vertebrate frugivores and higher in energy than unripe fruits, we hypothesised that volant vertebrate frugivores were selective in their fruit consumption and preferred ripe over unripe fruits (Corlett, 2011). Consequently, we predicted that the amount of fruit pulp eaten by flying foxes and birds would increase with increasing fruit ripeness. We then discussed the link between higher flying fox foraging activity outside forested areas during early night (Seegobin, Oleksy & Florens, 2022) and fruit damage differences in orchards between early and late night and the importance of quantifying such temporal damage. Previous studies have investigated temporal variation of flying fox movements at a broad island-wide scale in forested and cultivated areas (Oleksy et al., 2019; Seegobin, Oleksy & Florens, 2022), but none have investigated the temporal variation in fruit consumption by vertebrate frugivores in anthropogenic landscapes in Mauritius.

Materials & Methods

Site and species description

Mauritius (20°20′S; 57°34′E) is a volcanic island 7.8 M years old in the Indian Ocean about 900 km east of Madagascar (Fig. 1). It covers 1,865 km2, spanning about 60 × 40 km, has a maximum elevation of 828 m and is located within one of the world’s biodiversity hotspots (Myers et al., 2000). The mean annual temperature is 22 °C, and the mean annual rainfall varies from 800–4,000 mm (Staub, Stevens & Waylen, 2014). Habitat destruction due mainly to agriculture and urban development decreased its native forest cover to 4.4% (Hammond et al., 2015). Sugarcane plantations have occupied more than half of Mauritius (Nigel, Rughooputh & Boojhawon, 2015). However, sugarcane plantations are being converted to other crops including fruit (Ministry of Agro-Industry and Food Security of Mauritius, 2015). Lychee is one of the three main fruits produced on the island (Statistics Mauritius, 2017), and it has economic importance to both the local market (around 7,000 tonnes) and export industry (around 1,000 tonnes) (Fresh Plaza, 2016). However, data on its market value remain unavailable.

Figure 1 Mauritius in the Indian Ocean and location of the study sites.

The triangle at Morc. St André represents two nearby backyards. Morc. stands for Morcellement.

The Mauritian flying fox is a medium-sized fruit bat weighing 380–540 g and the last Pteropus species surviving on Mauritius following the extinction of two other species (Cheke & Hume, 2008). Its natural diet consists of leaves, nectar, flowers and mainly a wide variety of native fruits, making this species a keystone seed disseminator (Florens et al., 2017) and a potential pollinator (Nyhagen et al., 2005). Following the extinction of the two other Pteropus species, giant tortoises (Cylindraspis spp.), the dodo (Raphus cucullatus) and other large frugivores, P. niger has become the largest native seed disseminator of the Mascarenes (Hansen & Galetti, 2009; Heinen et al., 2023). It is known to be able to cross the whole island in a single night for foraging (Oleksy et al., 2019). The island has other native vertebrate frugivores, such as birds, comprising six passerine species and one parrot (Heinen et al., 2023). Apart from the Mauritius grey white-eye (Zosterops mauritianus), all the other native birds’ distribution is restricted to native forests (Jones, 1987; Safford, 1997). The Mauritius black bulbul (Hypsipetes olivaceus) is the main frugivore among native passerines but exotic birds (common myna and red-whiskered bulbul) competitively excludes it from non-native habitats (Safford, 1996). These exotic birds can move between native and secondary vegetation since they are highly mobile (Dulloo, Kell & Jones, 2002). Other alien frugivores include rats (R. rattus and R. norvegicus) and long-tailed macaques, but the red-whiskered bulbul is considered one of the most invasive species on Mauritius (Cheke, 1987).

Sampling and statistical analyses

With permission from the Ministry of Agro-Industry and Food Security, we sampled two lychee orchards, one at Calebasses (north, nine hectares) and one at Beaux Songes (west, seven hectares) and two backyard gardens in the north and one in the central upland region (Fig. 1). Twelve non-netted unharvested fruiting trees of ∼eight m height and 9–11 m canopy diameter were randomly selected in each orchard and a total of six trees of similar height and canopy diameter were selected in backyards. Random selection was done by assigning numbers to all non-netted fruiting trees in orchards and numbers were drawn using a random number generator. Backyard tree selection was dependent on the owner’s permission to enter the property at night and whether the tree would be left non-netted. Lychee season in Mauritius starts in September (fruit set), and the ripening stage spans November to end of December. The selected sites were visited regularly to monitor fruit growth and quadrats were set up before damage started, which was approximately six weeks after fruit set (following lychee ripeness stages (Wei et al., 2013; Chang et al., 2015)). Permanent quadrats of one m2 were placed at the four cardinal points, midway between the trunk and the canopy edge of each assessed tree, while slightly adjusting the placement of quadrats if the midpoint was located under foliage only.

Fruit ripening begins in the north region, and proceed southwards. Fruit damage started on 12 November 2022 in the north and 30 November 2022 in the west and central uplands and damage assessment lasted until no fruits remained in orchards or until fruits were harvested in backyards (28 November and 27 December 2022, respectively). We collected all the fruits falling into quadrats, identified the agents of frugivore damage (flying fox, bird, rat and macaque) using bite marks (Senar et al., 2016; Oleksy et al., 2021) (Fig. 2) and separated the amount of pulp eaten per fruit into five categories (0%, 25%, 50%, 75% and 100%) (Fig. S1). The corresponding ripeness stages of damaged fruits that still had their pericarp were also classified into percentages (characterised by change in pericarp colour from green; 0% ripe to red; 100% ripe) (Fig. S1). Sampling was done every six hours (06:00–12:00, 12:00–18:00, 18:00–00:00, 00:00–06:00) for 24-hour cycles. We recorded complete cycles three to five consecutive days per week for a total of 13, 11 and 15 days at Beaux Songes, Calebasses and backyards respectively throughout the fruiting season. The different bird species observed foraging on lychee during sampling were also noted. We estimated visitation frequencies of flying fox, bird, rat and macaque based on the presence of damage per tree, time and day. Data collected for backyard trees were focused on temporal quantification of flying fox damage only. One AudioMoth (recording 55 s every five minutes from 18:00–06:00) was used for two consecutive weeks in each orchard to record any deliberate disturbance made to deter flying foxes. We randomly selected 42 ripe fruits from a backyard tree and measured their weight (average 21.3 ± 2.3 g) to estimate fruit loss in kg.

Figure 2 Bite marks on lychee eaten by different species.

(A, B) Bats leave triangular shaped punctures on the pericarp, pulp or seed from their canines. (C) Birds (except parakeets) peck holes and feed on the pulp around the seed. (D) Parakeets tend to scrape the pulp from the top of the fruit or leave triangular marks. (E) Rats target the seed, leaving incisors marks. (F) Macaques leave incisors marks on the pericarp, pulp or seed. Degree of ripeness: unripe; 0% (D, E), 25% ripe (A) and 75% ripe (C). The scale is for images C and F only.

All statistical analyses were done in R (version 4.3.1) (R Core Team, 2023). Analyses were done for flying foxes and birds only as we had too few observations for other frugivores. We tested the hypothesized effects of time period (06:00–12:00 (morning), 12:00–18:00 (afternoon), 18:00–00:00 (evening) and 00:00–06:00 (night)) and successive sampled days on the amount of fruit eaten by flying foxes and birds per tree in every site (Beaux Songes and Calebasses for flying foxes and birds, and backyards for flying foxes) using generalized linear models (GLMMs) with negative binomial error distributions. We first fitted global models (using package glmmTMB (Magnusson et al., 2024)) with two-way interactions between day and site and between time period and site, to account for within-site variation in the relationships between number of fruits eaten, time period and day. For flying foxes, we only included two time periods (18:00–00:00 and 00:00–06:00), as they almost exclusively fed during these periods. For birds, we included all time slots, as they usually started feeding before 06:00 and finished feeding after 18:00. We included tree as random effect to account for correlation of repeated measurements per tree across time (Harrison, 2014). We evaluated model fit of all global models using residual diagnostic plots from package DHARMa (Hartig, 2022).

We used the global model for inference, as this also provides a balanced representation of statistically non-significant results. We first tested the global model against a null model with a likelihood ratio test (Forstmeier & Schielzeth, 2011). Next, we reported the model estimates and evaluated statistical significance of observed estimates using 95% confidence intervals (CIs) (Nakagawa & Cuthill, 2007). We considered evidence for an effect as weak, moderate and strong when the 90, 95 and 99% CIs did not overlap zero, respectively (Muff et al., 2022). For the two-way interaction between site and day in each GLMM, we calculated regression coefficients, standard errors (SEs), and CIs for the day slope for site, correcting for multiple comparisons using the Tukey method (package emmeans (Lenth et al., 2017)). For the two-way interaction between site and time period in each GLMM, we carried out post hoc contrast tests for the pairwise comparisons between time periods in every site (using package emmeans).

We also tested our hypothesised effects of stage of fruit ripeness on the proportion of pulp eaten by flying fox, parakeet or other birds in each orchard using a GLMM with ordered β distribution (and using individual fruits as sampling units). The ordered β distribution is similar to the zero–one-augmented β distribution (recommended for continuous proportions) (Douma & Weedon, 2019) but produces more accurate estimates and needs less processing time (Kubinec, 2023). In this GLMM, we included a three-way interaction between site, flying fox, parakeet or other birds and fruit ripeness stage and between site, flying fox, parakeet or other birds and day as fixed effects. We also included tree as a random effect to account for pseudoreplication. We used the same validation procedures as described for the previous GLMMs and used the global model for inference. We calculated regression coefficients, standard errors (SEs) and CIs for the slope of fruit ripeness stage for every species in every site and carried out post hoc contrast tests for the pairwise comparisons between predicted means in proportion of pulp eaten by different species for different stages of fruit ripeness in every site.

Results

Agents of fruit loss

Fruit loss by frugivores in orchards was caused by flying foxes (native) and birds, rats and macaques (all alien). We also accounted for fruit loss by other factors which included mainly fungal diseases, fruit cracking and natural fruit fall. The number of fallen fruits per square meter per day varied from 0–67 at Beaux Songes and 0–102 at Calebasses. At Calebasses and Beaux Songes respectively, fruit loss by flying foxes was 62.6–81.2% (n = 2,401) and 61.3–94.4% (n = 3,392); introduced birds was 9.8–30.4% (n = 711) and 2.4–26.7% (n = 289); and other factors was 1.2–12.2% (n = 191) and 1.3–12.0% (n = 183). Rat and macaque damage occurred only at Beaux Songes where it represented less than 1% (n = 5) damage. Only parakeet damage was distinguished from other birds. However, a total of four avian frugivores were observed feeding on lychees and they were all introduced species namely ring-necked parakeets, red-whiskered bulbuls, common mynas and village weavers (Ploceus cucullatus). The highest visitation frequency was by flying foxes (n = 446) and birds (n = 306) and the lowest was for rats (n = 3) and macaques (n = 1). Hence, flying foxes and birds were classified as regular visitors while rats and macaques were considered occasional visitors.

Temporal pattern of flying fox foraging behaviour

Of the fruits eaten by flying foxes, an average of 59% were eaten in the early night (54% Calebasses, 59% Beaux Songes, 68% backyards). The number of damaged fruits per tree (four m2) at Calebasses, Beaux Songes and backyards respectively varied from 30–280, 26–425 and 8–521 before midnight and 14–222, 20–226 and 7–196 after midnight. Although the estimated difference between early and late night was on average one (Calebasses), two (backyards) and three (Beaux Songes) fruits per tree per night, the GLMM analysis showed that the number of fruits consumed was significantly less in the late night than in the early night in Beaux Songes and backyards (Fig. 3, Table 1). At Calebasses, flying fox damage started with 44 fruits, peaked at 413 fruits (5th sampled day) after four days and dropped to 143 fruits on the last (11th) sampled day. The total number of fruits eaten at Beaux Songes started with 409 fruits on the first day, peaked at 573 fruits (4th sampled day) after four days and ended with 123 fruits on the 13th sampled day. They ate on average ∼1 fruit less per tree with increasing sampling days in both orchards but not in the backyards (Table 1, Fig. 4). The low R2marginal indicated that our predictors only explained 11% of variation in the amount of fruit eaten by flying foxes, meaning other factors unaccounted for probably considerably influenced the flying foxes’ feeding behaviour.

Figure 3 Fruit eaten by flying foxes before and after midnight.

Predicted means and 95% confidence intervals for total fruits eaten by flying foxes in the evening (18:00–00:00) and at night (00:00–06:00) in Beaux Songes (A), Calebasses (B) and backyards (C) as estimated by our Generalized Linear Mixed Models (GLMMs). The observations (number of fruits eaten per tree per day) used for the analysis have been plotted as individual points, with their respective tree identification number to illustrate the spread of the data.

Table 1 Effects of time period and day on the number of fruits eaten by flying foxes and birds.

The Generalized Linear Mixed Models (GLMMs) explaining the effects of time period (Morning: 06:00–12:00, Afternoon: 12:00–18:00, Evening: 18:00–00:00 and Night: 00:00–06:00) and day on the number of fruits eaten by flying foxes and birds on lychee trees in backyards and two orchards (Beaux Songes and Calebasses). For two-way interactions between time period and site, we provided regression coefficients, standard errors (SEs), 95% Confidence Intervals (CIs) and p-values for the pairwise comparisons between time periods for every level of site. For two-way interactions between day and site, we provided regression coefficients, SEs, CIs and p-values for the slope of day for every level of site. Regression coefficients are provided in the following format: Coefficient ± SE (CIlow, CIup). Effects for which the 90%, 95% or 99% CIs do not overlap zero are shown in bold. The final two rows contain marginal and conditional R2 values of the model. All estimates are on the response scale (number of fruit per tree).

	Response variable				
	Fruits eaten by flying foxes		Fruits eaten by birds		
Predictor/interaction	Coefficient ± SE (CIlow, CIup)	p-value	Coefficient ± SE (CIlow, CIup)	p-value	
Backyards: time period					
Night vs. Evening	−2.40 ± 1.26 (−4.86, 0.07)	0.06	–		
Beaux Songes: time period					
Morning vs. Afternoon	–		0.62 ± 0.15 (0.22, 1.01)	<0.01	
Morning vs. Evening	–		0.80 ± 0.17 (0.37, 1.23)	<0.01	
Morning vs. Night	–		0.28 ± 0.15 (−0.10, 0.67)	0.24	
Afternoon vs. Evening	–		0.18 ± 0.05 (0.05, 0.32)	<0.01	
Afternoon vs. Night	–		−0.34 ± 0.11 (−0.61, −0.06)	<0.01	
Night vs. Evening	−3.46 ± 1.37 (−6.14, −0.78)	<0.01	0.52 ± 0.11 (0.22, 0.81)	<0.01	
Calebasses: time period					
Morning vs. Afternoon	–		2.41 ± 0.63 (0.79, 4.03)	<0.01	
Morning vs. Evening	–		3.59 ± 0.72 (1.74, 5.45)	<0.01	
Morning vs. Night	–		3.54 ± 0.72 (−1.70, 5.38)	<0.01	
Afternoon vs. Evening	–		1.19 ± 0.26 (0.53, 1.85)	<0.01	
Afternoon vs. Night	–		1.14 ± 0.25 (0.49, 1.78)	<0.01	
Night vs. Evening	−1.02 ± 1.51 (−3.99, 1.95)	0.50	0.05 ± 0.03 (−0.01, 0.12)	0.18	
Successive sampling days					
Backyards	0.36 ± 0.33 (−0.28, 1.01)	0.13	–		
Beaux Songes	−1.34 ± 0.36 (−2.04, −0.65)	<0.01	0.02 ± 0.01 (0.00, 0.04)	0.06	
Calebasses	−1.09 ± 0.43 (−1.94, −0.25)	<0.01	0.11 ± 0.05 (0.01, 0.20)	0.03	
R2marginal	0.11		0.65		
R2conditional	0.36		0.71		

Figure 4 Seasonal variation of fruit eaten by flying foxes and birds.

Lines of best fit estimated by our two generalized linear mixed models (GLMMs) expressing the relationship between day and fruits eaten by flying foxes in the selected orchards (Beaux Songes and Calebasses) and backyards (top), and fruits eaten by birds in the selected orchards. The data points represent the number of fruits eaten per tree per day.

Temporal pattern of bird foraging behaviour

Fruits eaten by birds were on average 49% (n = 141) in the morning, 13% (n = 38) in the afternoon, 2% (n = 6) in the evening and 36% (n = 102) at night in Beaux Songes and 70% (n = 500), 28% (n = 199), <1% (n = 2) and 1% (n = 10) respectively at Calebasses. We found strong evidence that birds ate most fruits at night (00:00–06:00) and in the morning (06:00–12:00) in Beaux Songes and in the morning and afternoon (12:00–18:00) in Calebasses (Table 1, Fig. 5). The R2marginal indicated that our predictors explained a considerable amount of variation (65%) in the amount of fruit eaten by birds. The total number of fruits eaten at Beaux Songes started with 17 fruits on the first day, peaked at 39 fruits (9th sampled day) after 14 days and ended with 23 fruits on the 13th sampled day. At Calebasses, bird damage started with 11 fruits, increased to 90 fruits (5th sampled day) after four days and dropped to 66 fruits on the last (11th) sampled day. In contrast to flying foxes, we found weak to moderate evidence that birds ate slightly larger number of fruits toward the end of the study in both orchards (Table 1, Fig. 4). However, the relatively large SEs indicate that more observations were needed to obtain more precise estimates.

Figure 5 Fruit eaten by birds at different time periods.

Predicted means and 95% confidence intervals for total fruits eaten by birds for different time periods (Morning (M), Afternoon (A), Evening (E) and Night (N)) in the selected orchards as estimated by our GLMMs. The observations (number of fruits eaten per tree per day) that were used for the analysis have been plotted to illustrate the spread of the data.

Fruit ripeness and foraging behaviour of flying foxes and birds

Flying foxes ate between 20–28% more lychee pulp per fruit than parakeets at Beaux Songes and we found strong evidence of this trend at 0%, 25% and 50% ripened fruit (Table 2). In Beaux Songes and Calebasses, flying foxes ate 7% and 6% less fruit pulp than birds at the 25% ripened stage respectively, and it differed significantly at both sites. From unripe to 50% ripe fruits, parakeets ate 38–26% less fruit pulp than birds at Beaux Songes (strong evidence) and 7% less fruit pulp at 25% ripe fruit at Calebasses (weak evidence). Furthermore, flying foxes ate 39% and 42% more lychee pulp per fruit with an increase in fruit ripeness from unripe (0%) to fully ripe (100%) at Beaux Songes (strong evidence) and Calebasses (strong evidence) respectively (Table 2, Fig. 6). Parakeets ate 7% more fruit pulp with an increase in fruit ripeness at Beaux Songes and we found moderate evidence for this trend (Table 2, Fig. 6). With every increase in sampling day, flying foxes ate 1% more pulp per fruit at Beaux Songes (strong evidence) (Table 2). However, the small number of observations for parakeets and other birds for some levels of fruit ripeness resulted in large SEs, meaning additional observations were required to estimate the trends for parakeets and other birds more accurately. The low R2marginal indicated that other factors that we did not account for may explain a larger amount of variation in the proportion of fruit pulp eaten by different species.

Table 2 Effects of day and fruit ripeness on the proportion of lychee pulp eaten by flying foxes and birds.

The GLMM explaining the effects of day and fruit ripeness on the proportion of pulp eaten of lychee fruits by different species (flying foxes, parakeets and other birds) in the two orchards (Beaux Songes and Calebasses). For the three-way interaction between species, fruit ripeness and site, we provided regression coefficients, standard errors (SEs), 95% Confidence Intervals (CIs) and p-values for the pairwise comparisons between the different species for every level of site and fruit ripeness stages (0%, 25%, 50%, 75% and 100%). For both three-way interactions in our model (fruit ripeness, species and site, and day, species and site), we also provided regression coefficients, SEs, CIs and p-values for the slope of fruit ripeness and day for every level of site for every species. Regression coefficients are provided in the following format: Coefficient ± SE (CIlow, CIup). Effects for which the 90%, 95% or 99% CIs do not overlap zero are written in bold. The final two rows contain marginal and conditional R2 values of the model. All estimates are given in proportions.

	Sites				
Predictors/interactions	Beaux Songes		Calebasses		
	Coefficient ± SE (CIlow, CIup)	p-value	Coefficient ± SE (CIlow, CIup)	p-value	
Flying foxes–Parakeets					
ripeness 0%	0.24 ± 0.05 (0.12, 0.36)	<0.01	−0.05 ± 0.03 (−0.13, 0.03)	0.19	
ripeness 25%	0.27 ± 0.05 (0.16, 0.38)	<0.01	0.02 ± 0.02 (−0.04, 0.07)	0.99	
ripeness 50%	0.28 ± 0.05 (0.15, 0.40)	<0.01	0.09 ± 0.05 (−0.02, 0.19)	0.32	
ripeness 75%	0.25 ± 0.10 (0.02, 0.48)	0.11	0.16 ± 0.08 (−0.03, 0.34)	0.21	
ripeness 100%	0.20 ± 0.16 (−0.19, 0.58)	0.57	0.22 ± 0.11 (−0.04, 0.48)	0.20	
Flying foxes–Other birds					
ripeness 0%	−0.14 ± 0.07 (−0.3, 0.01)	0.10	−0.12 ± 0.06 (−0.26, 0.02)	0.10	
ripeness 25%	−0.07 ± 0.04 (−0.17, 0.03)	0.07	−0.06 ± 0.03 (−0.13, 0.01)	0.04	
ripeness 50%	0.01 ± 0.04 (−0.09, 0.12)	0.57	0.01 ± 0.04 (−0.10, 0.11)	0.92	
ripeness 75%	0.09 ± 0.07 (−0.07, 0.26)	0.99	0.08 ± 0.08 (−0.12, 0.27)	0.88	
ripeness 100%	0.17 ± 0.10 (−0.07, 0.40)	0.88	0.13 ± 0.12 (−0.15, 0.41)	0.73	
Parakeets–Other birds					
ripeness 0%	−0.38 ± 0.08 (−0.56, −0.20)	<0.01	−0.07 ± 0.07 (−0.22, 0.08)	0.60	
ripeness 25%	−0.34 ± 0.06 (−0.48, −0.19)	<0.01	−0.07 ± 0.03 (−0.16, 0.01)	0.08	
ripeness 50%	−0.26 ± 0.07 (−0.42, −0.10)	<0.01	−0.08 ± 0.06 (−0.21, 0.05)	0.33	
ripeness 75%	−0.16 ± 0.12 (−0.43, 0.11)	0.20	−0.08 ± 0.10 (−0.33, 0.16)	0.66	
ripeness 100%	−0.03 ± 0.19 (−0.47, 0.40)	0.79	−0.08 ± 0.15 (−0.44, 0.28)	0.80	
Fruit ripeness					
flying foxes	0.39 ± 0.05 (0.28, 0.50)	<0.01	0.42 ± 0.06 (0.30, 0.54)	<0.01	
parakeets	0.07 ± 0.14 (−0.20, 0.34)	0.02	0.16 ± 0.16 (−0.16, 0.48)	0.35	
other birds	0.31 ± 0.11 (0.09, 0.53)	0.12	0.14 ± 0.13 (−0.10, 0.39)	0.35	
Successive sampling days					
flying foxes	0.01 ± 0.00 (0.00, 0.01)	<0.01	0.00 ± 0.00 (0.00, 0.01)	0.18	
parakeets	0.02 ± 0.01 (0.00, 0.04)	0.90	0.01 ± 0.01 (−0.01, 0.03)	0.91	
other birds	0.00 ± 0.01 (−0.03, 0.03)	0.10	0.00 ± 0.01 (−0.02, 0.01)	0.21	
R2marginal	0.17				
R2conditional	0.19				

Figure 6 Relationship between fruit ripeness and foraging behaviour of flying foxes and birds.

The lines of best fit estimated by our GLMM expressing the relationship between the stage of fruit ripeness and proportion of pulp eaten per fruit by flying foxes, parakeets and other birds in two orchards (Beaux Songes and Calebasses). Boxplots of the raw data (the hinges represent the first and third quartiles and the upper and lower whiskers extend to the largest value up to 1.5 times the interquartile range (IQR) from the upper lower hinge) are also shown to illustrate the distribution of the data for different stages of fruit ripeness for each species in the two orchards. Number of observations per site, species and stage of fruit ripeness are indicated above every boxplot.

Deliberate disturbances in orchards

The AudioMoth recorded deliberate disturbances at both orchards. The number of events per six hours varied between 12-27 in the evening (18:00–00:00) and 1-30 at night (00:00–06:00) at Beaux Songes and 0-18 and 1-11 respectively at Calebasses. The duration of the sampled disturbances accounted for 3.0% (SD 1.4%) of the evening time period and 2.7% (SD 2.3%) of the night time period at Beaux Songes and 1.6% (SD 1.1%) and 1.0% (SD 1.0%) respectively at Calebasses. Five different types of disturbances were recorded and they were either noise-based deterrence (firecracker/gunshot, sound of stick hitting empty barrel, whistling and shouting) or smell-based (burning of leaves/branches/tyres) which could also be detected with the AudioMoth by the sound of crackling and sputtering during burning. However, since we were restricted to using a single AudioMoth throughout the fruiting season, we did not have enough data for more detailed analysis about the influence of such disturbances on foraging flying foxes.

Discussion

Impact of flying foxes and other frugivores on fruit loss

During our study, the percentage of damaged lychee attributed to flying foxes in orchards was 61–94% per tree. Given that we sampled an area of four square meter per tree, we estimated fruit loss by flying foxes between 19–272 kg of lychee per tree in orchards (based on an average tree diameter of 10 m (canopy area: 78.54 m2) and a fruit weight of 21.3 g). Oleksy et al. (2021) assessed fruit damage in 2015 at the same orchards as in our study to show the efficacy of netting trees and they recorded flying fox damage of 9% and 53% for non-netted lychee trees at Calebasses and Beaux Songes respectively. Both orchards covered between seven and nine hectares but Calebasses was divided into several smaller plots of approximately 0.4 hectares each. As a result, Calebasses had multiple owners, whereas Beaux Songes had only one. Hence, the reported differences between the two sites appear to have been influenced by the intensity of active deterrence, as the presence of multiple owners at Calebasses likely led to more pronounced deterrent measures. Furthermore, since 2009, the Mauritian Government has been providing subsidies to purchase nets for orchards and backyard gardens (Government of Mauritius, 2010). The higher (63% and 32% more at Calebasses and Beaux Songes respectively) flying fox damage recorded in our study suggested that the considerable increase in the use of nets in orchards during the last seven years may be displacing the damage by frugivores away from netted trees to concentrate them onto the fewer trees that remained unprotected. This would highlight the importance of protecting trees with methods proven as effective.

Bird damage was relatively low (3–68 kg per tree) and bird sonic deterrence (clapping, shouting and music) was occasionally used. Birds also seemed less affected by sonic deterrence, possibly explaining why a much larger amount of variation in bird foraging intensity was explained by time slot and day. Fruit loss by rats was negligible as they tend to hoard food (Bindra, 1948) and because rodenticides were used after fruit set. Even though macaques do visit Beaux Songes orchard (G Bhanda, pers. obs., 2018), their damage remained negligible. Macaques typically do not swallow lychee-sized seeds (Tsuji & Su, 2018), but may store them in their cheek pouches or carry them away in their hands over distances >20 m (Corlett & Lucas, 1990), particularly when disturbed by humans during crop raiding (from up to 200 m; R Reinegger, pers. obs., 2024). Hence, alternative methods for assessing crop raiding by macaques using camera traps paired with direct observations could be explored. Fruit loss by other factors was low because practices like watering adequately and frequently during fruiting to decrease the occurrence of fruit cracking (Marboh et al., 2017) and preventively spraying fungicides, were common. Furthermore, the maintenance of windbreaks around orchards could have reduced natural fruit fall. Apart from flying fox damage, damage by birds, alien mammals and other factors aligned with Oleksy et al. (2021).

Diel patterns

We confirmed that flying foxes followed a pattern while foraging as the amount of fruit eaten varied during the night. However, much of the variation in flying fox feeding intensity remained unexplained by foraging time or study period. Differences between trees explained a greater proportion of variation in feeding intensity. Hence, factors such as proximity to other fruiting trees and crop size could have affected fruit choice and frugivore’s foraging intensity (Manasse & Howe, 1983; Ortiz-Pulido, Albores-Barajas & Díaz, 2007). Nevertheless, since GLMMs residual plots indicated homoskedasticity and no other deviations, the effects of our predictors were still relevant. Hengjan et al. (2018) found a correlation between number of fruits dropped at different hours and frequency of flying fox visits. Hence, the amount of fruit eaten by flying foxes was considered as an indicator for its density. The higher flying fox density recorded in early night at Beaux Songes orchard and undisturbed backyards suggested that they were targeting familiar foraging patches with energy and water-rich foods. Flying foxes in Mauritius demonstrated seasonal movement patterns depending on food availability (Oleksy et al., 2019) with more flying foxes foraging outside natural forest areas during commercial fruiting seasons (Seegobin, Oleksy & Florens, 2022). Studying the black flying fox (P. alecto), Markus & Hall (2004) showed its tendency to fly directly towards known foraging sites at nightfall. Another study on Egyptian fruit bats (Rousettus aegyptiacus) suggested that experienced bats time their visits with an understanding of tree phenology and showed that bats that have not eaten much and did not drink for 12 h would leave their colony early to target water-rich fruits compared to late-leaving bats seeking protein-rich fruits (Harten et al., 2024).

Non-netted orchard lychee trees sustained high fruit losses to flying foxes despite being subjected to some degree of sound and smoke deterrence. These deterrence practices are common in most Mauritian orchards. Orchard owners would either camp there during fruiting season or employ people to do so to actively deter flying foxes and thieves at night. Despite the higher flying fox damage recorded before compared to after midnight at Calebasses, the difference was not statistically significant. Since both orchards were subjected to active deterrence, it seemed that flying foxes at Calebasses could have been more affected as the difference in the duration of disturbance before and after midnight was slightly higher than at Beaux Songes. Disturbances may have varied in intensity between different parts of the orchards too, further explaining between-tree differences in flying fox foraging intensity.

Using active deterrents only during early nights would protect on average ≤1 kg more lychees per tree per night. In some cases, it could protect up to 42 kg more lychees per tree per night, but in other cases, it could result in a maximum loss of 20 kg. Additionally, animals tend to shift their activity with human disturbance (Gaynor et al., 2018; Lee et al., 2024). Consequently, deterring flying foxes only before midnight could lead to more flying foxes foraging in orchards after midnight. Although effective, as a result of the short term renting of orchard trees for exploitation that precludes cost-effective multi-year netting, fruit sellers find netting trees to be more costly, labor-intensive, time-consuming, than it ought to be, and point out that net supplies are often limited, especially during the fruiting season. For example, adopting active deterrence method can cost 2.5 times less than netting a lychee orchard of one hectare (∼180 trees) (orchard owner, pers. comm., 2024). Hence, for orchard owners relying only on active deterrence to protect trees from flying fox damage, it is recommended to protect the orchard during the whole night. While deliberate human disturbances involving the use of smoke or sound has been reported in Australia and India to deter frugivorous bats (Srinivasulu & Srinivasulu, 2001; Bicknell, 2002), their effectiveness in decreasing bat damage in orchards remained to be studied. No study demonstrating the effectiveness of these commonly used methods has been published in Mauritius.

Our study was done in summer with sunrise around 40 min before 06:00 and sunset 35 min after 18:00, explaining the record of bird damage at “night” (approximated to 18:00–06:00 in our study). Studies that modeled the optimal foraging behaviour of birds by including predation and starvation risks often predicted a bimodal feeding pattern with early morning and late evening peaks (Bednekoff & Houston, 1994; McNamara, Houston & Lima, 1994). Since our study was limited to diurnal data collected before and after noon, we could not investigate this bimodality. However, we found an early morning foraging peak followed by a decreasing rate of foraging throughout the day at both orchards. This peak could be explained by unpredictable food sources (influenced by weather changes and interruptions by competitors and predators) and a higher starvation risk in early morning pushing the birds to replenish energy reserves exhausted overnight (Bednekoff & Houston, 1994). When feeding is uninterrupted, birds are expected to decrease their foraging activity the rest of the day, feeding to maintain energy reserves and low predation risk (Bednekoff & Houston, 1994; McNamara, Houston & Lima, 1994). At Calebasses, foraging was significantly higher approximately one hour after sunrise. At Beaux Songes, however, foraging was significantly higher from sunrise possibly due to the residing large colony of village weaver nesting on ironwood trees (Casuarina equisetifolia) found inside the orchard. This is an additional and fourth alien bird species feeding on lychee in Mauritius compared to Oleksy et al. (2021).

Relationship between seasonality, fruit ripeness and foraging

Flying foxes and passerine birds at Beaux Songes ate more lychee pulp from ripeness stages 0% to 50% compared to parakeets. Birds at both orchards ate more pulp from 25% ripe lychees compared to flying foxes. Flying foxes ate more fruit pulp with increasing ripeness at both orchards and more fruit pulp towards the end of the study at Beaux Songes. This trend was not apparent at Calebasses as, for most of the assessed trees, all fruits were eaten before they ripened. While flying foxes are known to prefer ripe fruits (Luft, Curio & Tacud, 2003), we could not assess this preference in our study as we did not account for the availability of fruits of different ripeness stages on trees. However, consistent with previous findings, we could speculate that they favour ripe fruits (Krivek et al., 2020; Reinegger et al., 2021). Flying fox damage began in the seventh week after fruit set, when fruits were still unripe, with most damage occurring between the unripe and 25% ripened stages, during which they consumed little fruit pulp. This behaviour is common in foraging animals as they tend to become more choosy during high fruit availability when the probability of obtaining fruits of higher quality increases (Pyke, 1984; Janson, 1996; Whitehead, Quesada & Bowers, 2016). Hence, netting trees at latest in the sixth week after fruit set is advisable, a practice that is rare among most orchard owners who often end up netting their trees when fruits start ripening (G Bhanda, pers. obs., 2018–2022).

Ring-necked parakeets also ate larger proportions of pulp with increasing fruit ripeness in Beaux Songes, indicating that they could also target ripe lychee pulp. This suggested that they could be more frugivorous than granivorous (Shivambu, Shivambu & Downs, 2021). Consequently, like flying foxes, ring-necked parakeets were also more wasteful when feeding on unripe fruits, similar to findings by Sebastián-González et al. (2019), as they would only bite the fruit or partially consume it before discarding the rest. However, additional observations were needed to confirm how strong this trend was among parakeets in Mauritius because most fruits had been consumed when unripe leaving few ripe ones. For both orchards, frugivores ate less fruits towards the end of the fruiting period because fewer fruits remained on trees as they had already started depleting fruits from the beginning.

Conclusions

Our study emphasized the need to deploy effective crop protection strategies like netting earlier during the fruiting season (before the seventh week after fruit set), which falls between the first week of October and first week of November (taking into consideration intra and inter annual range for fruit set). This could be facilitated by ensuring an early and adequate supply of bird nets in the local market. We also highlighted the importance of assessing the feasibility of proposed management strategies related to HWC. Despite having more flying fox damage before compared to after midnight, the difference in fruit loss between the two time periods showed that fruit growers would still face considerable losses if they concentrated the use of active deterrence during early nights despite more damage to fruits happening then. Although our study was limited to two orchards, the overall nocturnal deterrent practices at these sites were consistent with the practice widely adopted throughout the island (G Bhanda, pers. obs., 2018–2022). A future, broader, island-wide investigation may reveal the influence that factors like orchard size, proximity to native forests or roost sites, or varying densities and diversity of other frugivores, may have on flying fox foraging patterns. Finally, this study opens the way to assessing the relative effectiveness of various active (e.g., reactive to frugivore presence like using firecrackers) or passive fruit crop protection methods (e.g., use of different types of netting) in attempt to help alleviate the HWC.

Supplemental Information

Supplemental Information 1 Ripeness stages of lychee and categories of amount of lychee pulp eaten by frugivores

Lychee at different ripeness stages and different categories of amount of lychee pulp eaten by flying foxes and birds. (A) Lychee ripeness from unripe (left; 0% ripe) to fully ripe (right; 100% ripe). (B) Bite marks on lychee showing that the fruit was damaged but not eaten (0% pulp eaten). (C) >0%–25% pulp eaten. (D) >25%–50% pulp eaten. (E) >50%–75% pulp eaten. (F) Fully eaten (>75%–100%).

Supplemental Information 2 Frugivore foraging in orchards

Supplemental Information 3 Frugivore foraging in backyard gardens

Supplemental Information 4 Lychee ripeness stages and amount of fruit flesh eaten by frugivores in orchards

Supplemental Information 5 R codes for Data S1, S2 and S3

We thank orchard owners (Mr Jay Khaidoo and Mr Hurry Lutchmun) for allowing use of their trees for this study. Yogeeta Devi Luchoomun from the Food and Agricultural Research and Extension Institute (FAREI) of the Ministry of Agro-Industry and Food Security (MoA-FS) provided data on dates of flowering and fruit set of lychee. Thalia Klotz helped process the AudioMoth recordings. We thank all the volunteers for their help in data collection on the field as well as several staff members of the University of Mauritius for their various support. We also thank the editor, Donald Kramer and Paul Racey and a second anonymous reviewer for their valuable comments.

Additional Information and Declarations

Competing Interests

Author Contributions

Data Availability

The authors declare there are no competing interests.

Geetika Bhanda conceived and designed the experiments, performed the experiments, analyzed the data, prepared figures and/or tables, authored or reviewed drafts of the article, and approved the final draft.

Ryszard Z. Oleksy conceived and designed the experiments, authored or reviewed drafts of the article, and approved the final draft.

Raphaël D. Reinegger analyzed the data, prepared figures and/or tables, authored or reviewed drafts of the article, and approved the final draft.

Cláudia Baider performed the experiments, authored or reviewed drafts of the article, and approved the final draft.

F.B. Vincent Florens conceived and designed the experiments, authored or reviewed drafts of the article, and approved the final draft.

The following information was supplied regarding data availability:

The raw data and code are available in the Supplementary Files.

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
