# Peer review of "A study of diel and seasonal patterns of loss of commercial lychee fruits to vertebrate frugivores: implications for mitigating a human-wildlife conflict"

_PeerJ, doi:10.7717/peerj.19269_

## Round 0.1 · original submission · Major Revisions

Overview: This study examined the amount and sources of fruit loss from lychee trees in commercial orchards and private yards in Mauritius by examining the number and appearance of damaged fruits in quadrats placed under the trees over the production period. The majority of loss was due to an endemic, endangered fruit bat with a substantial amount due to birds and very little to other sources including macaques, rats and other sources such as fungus. Quadrats placed under the trees in the orchards were sampled at 00:00, 06:00, 12:00, and 18:00 to examine diel patterns of loss from the different sources. For bats, fruit loss occurred only between 18:00 and 06:00, and there was significantly more fruit loss before than after midnight in one orchard and in yards, but only a trend in the other yard, and fruit loss tended to decline over time as the fruit ripened. For birds, fruit loss occurred in all four time periods, with the highest loss before 06:00 declining successively in the other periods, though not all the differences were statistically significant, and there was a very slight increase in loss over time. There was also information on proportion of fruit consumed for different species of frugivores at different ripeness stages.

Both reviewers considered the manuscript a valid and useful contribution to the literature. They suggested only a few minor revisions. However, my reading brought up numerous issues related to clarity and completeness not mentioned by the reviewers. You may regard these as a third review, i.e., make appropriate changes if the suggestion is valid or clearly explain why you do not find it valid. I have also provided numerous annotations on the pdf minor issues of conciseness, clarity, and grammatical correctness. You do not need to mention each of these annotated comments in your rebuttal unless you disagree and have not made the suggested changes.

Editor’s Comments
General concerns
• The manuscript appears somewhat unbalanced. The data collection considered all species of frugivores, and the data analysis and figures gave equal weight to birds and bats. However, the Introduction is largely focussed on bats and the title doesn’t even mention birds. The manuscript could use a reorganization that matches the Introduction to what you actually did.
• The manuscript can be hard to follow because it does not maintain a consistent order of questions. A reorganization is required so that the topics introduced in the Introduction, the objectives of the study, the methods, the results and the discussion all follow the same order.
• Although the manuscript takes an applied perspective, the application seems rather vague. I don’t really see how one could apply the timing of frugivory in broad 6-hour blocks to reduce damage. As a behavioral ecologist, it seems obvious that if one block of time receives more disturbance animals will forage at another (perhaps with the restriction of consistent diurnal and nocturnal activity, although there is good evidence of bears becoming more nocturnal when more disturbed by people). Clarify this by adding any evidence from previous studies of other species that detecting general activity times can be used successfully for reducing damage. If there is no such evidence, make it clear what you are proposing and that this is only the first step in using deterrence as a means of reducing conflict.
• There are frequent unjustified tense changes. Please reread the manuscript carefully so that consistent tenses are used.
• It would help the reader if you either indented paragraphs or skipped a space between them to make the organization clearer.
• After introducing each species, refer to it by a consistent term throughout the manuscript. I noticed that you used P. niger, bat, and flying fox to refer to the same species and that you sometimes referred to the birds by their common name and sometimes scientific name.

Specific comments

Title: The title is very long, incomplete because there is no mention of birds which are a significant part of the contents, and misleading because there is minimal examination of mitigation strategies. Consider something like ‘Loss of lychee fruits to vertebrate frugivores in Mauritius, diel and seasonal patterns’.

Abstract:
The Abstract is long and wordy. Revise to remove redundancies, focus on what the study actually found provide more quantitative information (as outlined in comments on Results below).
L26. Provide scientific name and correct common name as well as generic term ‘flying fox’.
L38. Specify the birds, with common and scientific names
L44. I don’t recall this point being made in Methods (study design) or brought into the Discussion.

Introduction:
The Introduction should provide the reader with concise but clear information about all topics considered goals of the study so that when the objectives are mentioned their relevance is clear. Your Introduction did not provide information about frugivores other than bats that cause loss of fruit crops, what is known about the diel and seasonal timing of fruit loss from fruit bats and other frugivores, or what is known about preferences and exploitation of different stages of fruit ripeness by these frugivores.
L73. In one sentence or as part of the first sentence, introduce this group of organisms, for example stating that they are frugivorous/herbivorous bats and providing the names of relevant families and/or genera.
L93. Ae these percentages the percent of fruit damaged or percent of growers reporting damage or something else? Presumably, they are based only on growers who did not use nets as protections?
L99. Do you mean ‘reducing fruit losses’ rather than a broader increase in production?
L100-103. This sentence should come after you identify goals of your study.
L104. You have suddenly switched from referring to P. niger to frugivores in general. You need a paragraph here to state that in general other species have also been recognized as important threats to fruit crops and that P. niger is not the only frugivore on the island. You could also indicate why it is important to know which frugivore(s) are causing the loss of fruit.
L104-105. You need to follow this topic sentence with a paragraph providing a background to how studies of diel patterns, seasonal patterns, and relationship to fruit ripeness have provided useful mitigation strategies in other contexts. If there are no examples from the literature, you need a clear statement about why these may be important in your system.
L108. ‘Patterns’ is too vague; be specific about what patterns.
L114. To hypothesize and predict are not the same thing. Do you mean hypothesize here as you stated in previous statements?
L115. I don’t see how this is different from L112
L115. Do you only hypothesize a difference, not the direction of the difference?
L117-118. You don’t test any direct hypotheses about decision-making or cues. Revise to bring the hypothesis in line with what you actually tested. You need to explain how patterns of crop maturation on the tree (seasonal effect) and patterns of percent consumption per fruit relate to the concept of increasing consumption of riper fruits.
L120. I don’t see how this is different from L116-117. You have not clearly differentiated the purpose of examining seasonal effects from the purpose of examining ripeness within a sample.

Methods
L150. If selection was truly random, provide a brief indication of the randomization method. If no random protocol was used, the selection should be described as ‘haphazard’.
L150. Were the backyard trees similar in dimensions to those in orchards. If so, indicate; if not, indicate dimensions.
L151. Where were the quadrats placed in relation to distance between the trunk and edge of the canopy?
L152ff. This information is incomplete and out of logical order. Remember that a reader should be able to repeat the study from the information in the Methods. Before referring to sampling dates, provide information on the start and end of the lychee fruiting season and make it clear to what extent the ripening is synchronized or not because these have an important bearing on understanding and interpreting your results. Keep the order of methods parallel to the objectives, identifying frugivore species, counting fruits damaged by each species, diel sampling, seasonal sampling, and ripeness measures. Possibly, ripeness and proportion of fruit consumed would go more logically with the identification of species causing damage since it was presumably done at the same time.
• Were the fruits all removed from the quadrat?
• Were the fruit measures done at the time they were picked up or later?
• Presumably, for the birds, some of the identification came from direct observations rather than fruit damage. How systematic was this and where and when was it done?
L152-153. Readers need more information about the species identification.
• Are all damaged fruits dropped or could some be totally consumed or carried away from the tree?
• How did you verify your ability to identify species? Were there any cases where the species was ambiguous? How often did this occur? Is there previous literature on this?
L154. Why did you select these times for sampling? How did they relate to the light cycle and activity? Why only these times? It strikes me that the sampling times were not ideal because it was light enough for birds to forage before the first sample so that the midnight to 06:00 sample includes both foraging and non-foraging time which is not ideal for your analysis of proportion of damage in different time periods. In addition, most studies of diel foraging patterns use a much finer scale such as hourly or a time frame adjusted to the light (or other) cycle so I think you need to explain this sampling plan since you put considerable emphasis on the diel pattern. The relevance of before and after midnight for bats could have been explained in the Introduction so that this timing makes sense to readers.
L155. I think it will be clearer for readers to use the term ‘day’ rather than ‘cycles’ here and throughout the results, figures and tables.
• The data are presented in terms of number of days, so specify here how many days were actually measured. The dates provided don’t seem to be an even number of weeks, so it is not clear whether 3 days would have been sampled on the last partial week. I calculated 6.5 weeks between Nov 12 and Dec 27. This could imply 18 to 21 days. Fig. 3 seems to show variation between sites of 11-15 days for bats and 11 to 13 days for birds. Neither the variation between sites nor the total number of days seems to match the description in your methods.
• Also, this section seems to specify both a terminal date and a terminal criterion (no fruit remaining), but you need to make clear which determined the decision to stop. Although I have no experience with lychee trees, from my experience with other fruiting trees, it is hard to imagine that the last fruit was removed from all the trees on exactly the same date. If few to no fruits were left by Dec 27 so you decided to stop sampling, it would be clearer to state something like ‘We started sampling on 12 Nov when [give an idea of the state of the crop on this date] and ended on 27 Dec when there were no (few?) fruits remaining on any of the trees. We recorded complete cycles three days per week for a total of X days.’
L156-159. This information fits logically with the species identification. Revise this to have separate sentences for percent removed and ripeness. Figure S1 is nicely presented but needs a complete caption. You have four categories but show 5 examples for each, so it is unclear. Presumably, the percent categories are ranges; 0% must be 0 – 12% or something, not only zero; make this clear. Remember, another researcher should be able to repeat the study.
L160. Why?
L166. Consider using ‘day’ and ‘time period’ or just ‘time’ throughout the manuscript to be clearer to readers.
L166. Use a consistent order for variables. Your results address time before days, so it seems reasonable to use this order of presentation here and below.
L166. Consider whether it would be clearer and more concise to designate terms for each time period instead of writing out the times each time you need to refer to them. For example, I wonder if you could call them simply, morning, afternoon, evening, and night, explaining to the reader once how these link to actual times.
L166. For the effect of day (what you call cycle), did you use successive numbers for each sampled day or use the number of days since the start of the study? This could make a difference in the analysis because of the gaps between sampling days. Make this clear and provide a justification for your choice if you think it necessary.
L168. Consider using ‘taxa’ when referring to bats and birds in general, and ‘species’ when referring to particular species of birds. ‘Animals’ is a bit unclear and could refer to individual animals.

Results
L207. Since you are presenting overall percent losses, it would seem reasonable to first present overall total losses on some comparable and easy to understand scale, perhaps number of dropped fruits per square meter over the duration of the study. When you address time period, then give totals per tree over the duration for each time period. When you address day (season), you could give the number per day from the beginning to end of the study period. It would be valuable in terms of applied significance to relate these numbers to the number of fruits available on the tree, if you have any measure of that. Even a qualitative trend might be helpful for readers to understand the impact of the frugivores. An additional useful piece of information would be how to scale the quadrat values up to the scale of losses per tree. What is the area of fruit production which is sampled by the 4 sq m per tree?
L208. Assuming that the n-values on L214-215 are total number of damaged fruits, it would be clearer to give these numbers adjacent to the percentages. Clarify what the +/- represent and at what scale – sites, days, time. If it is sites, probably range would be more relevant than SD.
L210. Revise sentence to say damage from these sources occurred only at Beaux Songes where it represented x% of damage in this orchard.
L211. The way this is presented, it is not clear whether all avian frugivores were introduced species or whether you didn’t count native frugivores. Clarify.
L218. You should provide more information on the effect size instead of focussing exclusively on significance of the difference. It is at least as important to know how big a difference there is between early night and late night as whether that difference is statistically significant. Given the similarity of the horizontal lines in Fig. 3, it seems to me that the differences are not great, even though some are statistically significant. I suggest starting this section of the results with something like the following ‘Of the fruits eaten by bats, an average of A% were eaten in the early night (B% Calabasses, C%Beaux Songes, D% backyards)(Fig. 3). Although 95%CIs overlapped extensively between early and late night periods (Fig. 3), the GLMM analysis showed that the number of fruits consumed was significantly less in the late night than in the early night in Beaux Songes and backyards (Table 1).’ Take a similar approach to the seasonal trend with a clear indication of the effect size.
Figure 3. This figure and its caption need to be clarified. For the diel effect on bats, the numerous points overlap extensively and do not add to understanding the figure because the vertical axis is so great that the differences between the means cannot be clearly perceived. After several readings, I think I understand that the mean and CI are based on the GLMM but the individual points represent the actual data, but this is not clearly stated. Potentially, each point could represent the sum for a single quadrat, a single tree, all trees at one site, all trees at all sites on a single day or for some combination of days. The reader needs to know that and how that changes for the daily and seasonal figures. While the units of n are adequate for the axis label, clarification in the caption is needed. You need to find a way to show the central tendency more clearly. Consider a bar graph showing the mean with a vertical line for the 95% CI. Are you certain that the significant differences you described occur despite the extremely large overlap of the CIs? Note also that the legend takes up space on the figure and forces the graph panels to be reduced in size to fit on the page. I suggest using a code for the time periods under each bar, explained in the caption. This could be A, B, C, D or something that more clearly stands for the time, e.g., M, A, E, N for morning, afternoon, evening, night. You could also provide information on the p-values which are not in Table 1 by using brackets between the bars to show the significance of differences for time of day and a simple p-value above each graph to show the significance of the seasonal trend. Be sure that the caption is clear and complete. For example, now you do not state what the points represent or what the sample size is.
Table 1. Consider whether names for the time periods would be easier for readers as suggested above. Where there are no comparisons, put the dashes in the middle of the column. Explain what a positive and negative sign on the coefficients means. Consider whether you can find a way to show the p-values in the table because the reader doesn’t know whether the lack of overlap with zero indicated by bold font is for 90, 95 or 99%. (Similar issues in Table 2).
Fig. 4. Boxplots can be structured in different ways. Be explicit about the meaning of the horizontal line, box, vertical line and dots. Correct the y-axis label; the numbers show percent, not proportion.
L224-226. This is not clear. The values presented seem to indicate less fruit used in late night. If there is an average trend with small effect size, it would not be surprising to have some exceptions.
L226. Clarify throughout the manuscript what your measure is. Is fruits per tree an extrapolation or is it really fruits per 4 meters squared?
L227. It is not clear what ‘proportion duration’ means. Why not use number of events per 6 hours?
L227. Put spaces around the +/- signs and also indicate what the variation refers to (SE, SD, CI, range?)
L236. The section on birds presents data out of consistent order. Follow the above suggestions with an initial indication of effect sizes, time before season before presenting the statistical support or lack of support for differences.
L243ff. This section and its relevance are unclear. The Introduction did not prepare the reader for the questions that this section is attempting to answer.
• Note that you cannot assess cause and effect (as in the subhead) from a correlational study. After clarifying the question, you should present this as associations/correlations.
• Following the advice on the previous sub-sections, you should clearly indicate effect sizes before presenting statistical significance.
L259. This information is out of place. It is not clear what its relevance is. Could it go in the methods when you describe lychee trees and fruits?

Discussion
• The Discussion requires a major reorganization. You should address each of the objectives of your study, in the order used in the Introduction, Methods, and Results, considering both the strengths and weaknesses of your inferences, based on issues such as sample sizes, effect sizes, statistical significance, and any limitations from the study design. Then relate your findings to previous research, making clear what findings are novel and which provide additional support to previous studies.
• It is important to consider the availability of different ripeness stages. If you do not have this information, any discussion of preferences is not appropriate, although you might be able to compare species foraging at the same sites and same times to infer differences in preferences of different species. Furthermore, availability of alternative types of foods could affect consumption. Preference (L264) cannot be based on simple results but is an inference requiring careful argument.
L272. Why remarkable?
L276. Clearly define what you mean by ‘wasteful’.
L264. Could they really infer ‘an understanding of tree phenology’ rather than a simple response to seasonal variation? What do you (or they) mean by ‘understanding’?

Applications
L317. ‘High losses’ presumably refers to a proportion of total fruit lost. You have not made this inference. For example, if trees produce several thousand fruits, 100 fruits damaged by bats would not be considered a high loss. More losses to bats than to birds is not an indication of high losses per se. You need to provide a synthesis, for example, an estimate of the total loss per tree per year related to the total production per tree. This would be very useful in the applied context, even if it was a very rough estimate. Some of this calculation might be more appropriate in the Results than in the Discussion, and it could be explicitly one of your objectives. L238 seems to imply a huge effect of deterrence. We need to see an explicit presentation of the calculations leading to this inference.
L327. I think that the advice about deterrence needs to take a more cautious approach here and in the Abstract. The behavioral ecology literature provides much evidence that animals avoid disturbance by shifting their activity times. It is possible that bats disturbed early in the night will simply shift to later night. It would be appropriate to find cases where this type of approach worked for bats (or any other species) and to call for experimental studies to test its effectiveness. This approach seems very time-demanding in relation to nets and probably much less effective; perhaps you should discuss briefly the economic or other reasons why it might be preferred.
L367. This section seems quite redundant. Perhaps you should focus on additional research needs to study the relevant ecology and effectiveness of mitigation methods.

·

Basic reporting

The paper ,meets all four criteria although I have some suggestions regarding English expression (see below)

Experimental design

No comment

Validity of the findings

No comment

Additional comments

Some suggested improvements to English:
Line (l) 30: 24 hour (not hours)
l 36: bats (plural)
l 41: forms (plural)
l 46/47: the latter
l 52 : concentrating earlier in the season.
Which season? The fruiting season?
l 52: for bats (plural)
early morning for birds
l 159: data (plural) which were focused
l 209: 'at' is redundant
l 248/349: sonic not sonor
l 363: birds'

Reviewer 2 ·

Basic reporting

The present manuscript, entitled "Extent of damage to commercial fruits by a Pteropus species varies through a single night and through the fruit season, offering new prospects to mitigate human-wildlife conflicts" and authored by Bhanda et al., is a relatively simple yet elegant evaluation of the fine-tuned temporal patterns of lychee damage by native and alien frugivorous vertebrates in Mauritius, with a particular emphasis on the Greater Mascarene Flying Fox (currently classified as IUCN Endangered, largely due to large-scale culls stemming from conflicts with fruit producers). The key findings reiterate that, compared with other frugivores, the threatened flying fox does indeed have a considerable impact on fruit damage, further emphasizing the need for evidence-based measures to mitigate conflicts with fruit growers.

Experimental design

Notwithstanding the relatively small number of surveyed orchards and backyard gardens, the authors manage to provide substantial - although somewhat site-specific - evidence of relevant within-night (bats) and within-day (birds) temporal dynamics of fruit damage. These findings, alongside the observation of increasing fruit damage throughout the fruiting season, are novel and can aid in devising more fine-tuned strategies to alleviate human-bat conflicts.

Validity of the findings

The manuscript is very well written, the field and statistical methods are suitable for the questions being raised, and the introduction links well with the discussion and conclusion. These are well supported by the results and contextualized according to relevant literature.

Additional comments

My main criticisms (intended to be constructive) are:

1. The potential implications of the small number of localities investigated should be better discussed.

2. The organization of the discussion could be improved with the use of more descriptive subtitles (e.g., emphasizing differences between the impacts of bats and other frugivores, or contrasting within day/night patterns vs. dynamics throughout the fruit season).

This work offers valuable and novel contributions to the literature and has the potential to positively influence the management of bat-farmer conflict. In addition to the above criticisms, I have only minor comments outlined below:

• Line 114: Why do you predict that the amount of fruit damage would differ before and after midnight?

• Line 148: Consider adding the number of backyard gardens surveyed.

• Line 149: “Nine” (in “nine-11 m”) should be numeric (“9”), as it is associated with a unit.

• Line 336: Consider elaborating on the reasons behind the drastic differences in reported fruit damage for non-netted lychee trees in both regions.

• Table 1: Consider reminding the reader of the meaning of “cycle” (as done at the end of the legend for Fig. 3), as the term is not very intuitive without context.

• Table 2: Some CIs currently highlighted in bold marginally overlap with zero. Please confirm if they should indeed be highlighted as significant.

• Fig. 2: Consider adding the degree of ripeness of the fruits displayed in the images to the legend. This would help unfamiliar readers visualize the scale used for quantifying ripeness (mentioned, e.g., in Fig. 4).

---

## Round 0.2 · accepted · Accept

The revisions were appropriate and carefully done, and the rebuttal was thorough. Following an email discussion of a small number of outstanding clarifications, I now consider the manuscript ready for publication.